# BaWA: Automatic Optimizing Pruning Metric for Large Language Models with Balanced Weight and Activation

**Lian Liu** [1 2 3 4]  **Xiandong Zhao** [1]  **Guanchen Li** [1]  **Dong Li** [1]  **Mengdi Wang** [2 3]  **Yinhe Han** [2 3]  **Xiaowei Li** [2 4]  **Ying Wang** [2 3]

## Abstract

One-shot post-training pruning enhances the deployment of billion-scale large language models (LLMs), with the pruning metric playing a pivotal role in determining which weights to remove. However, existing metrics underperform due to their reliance on a simple symbolic combination of weights and activations, overlooking imbalanced weight magnitudes and the disproportionate influence of activation outliers. To overcome these limitations, we introduce **BaWA**, a novel pruning metric that systematically Balances Weight and Activation distributions for more effective pruning. BaWA introduces two key innovations: **magnitude normalization**, which mitigates weight imbalance across channels for fairer pruning decisions, and **outlier regularization**, which reduces the impact of activation outliers, ensuring more appropriate channel prioritization. To further enhance its effectiveness, BaWA incorporates an efficient and automatic framework for optimizing normalization and regularization hyperparameters. Extensive experiments validate BaWA as a state-of-the-art (SOTA) pruning metric. For instance, applying BaWA to induce 2:4 sparsity in Mistral-7B reduces perplexity in language comprehension by 2.49 and improves average downstream task accuracy by 3.08%, outperforming the previous SOTA method Wanda.

## 1. Introduction

The remarkable performance of large language models (LLMs)(qwe, 2024; Dettmers et al., 2022; Touvron et al., 2023a;b) has revolutionized AI research, demonstrating exceptional capabilities across diverse and complex tasks. However, their vast parameter sizes, often in the billions, pose significant hardware constraints, limiting their practical deployment, especially in resource-constrained environments (Liu et al., 2025b). To mitigate this challenge, model compression techniques such as quantization(Dettmers et al., 2022; Xiao et al., 2023; Lin et al., 2023; Yuan et al., 2023; Frantar et al., 2023; Shao et al., 2023; Liu et al., 2024; 2025a) and pruning (Li et al., 2023a; Kwon et al., 2022; Frantar & Alistarh, 2022; Xia et al., 2023) have been explored to reduce resource demands while maintaining performance. This paper focuses on unstructured pruning, which compresses models by zeroing out weights and benefits from sufficient hardware acceleration supports.

Due to the scale of LLMs, one-shot post-training pruning has gained attention for its ability to efficiently remove weights without relying on gradient backpropagation or fine-tuning (Frantar & Alistarh, 2023; Sun et al., 2023; Zhang et al., 2024; Dong et al.). While effective, existing methods remain constrained by suboptimal pruning metrics. At the cost of additional mask training, MaskLLM (Fang et al.) demonstrated that a well-designed pruning metric enables weight sparsity with minimal performance loss. However, existing one-shot LLM pruning methods, such as Wanda (Sun et al., 2023) and Pruner-Zero (Dong et al.), rely on a simple symbolic combination of weights and activations to determine pruning decisions. While intuitive, they overlook the heterogeneity in weights and activations, leading to suboptimal pruning masks and significant performance degradation in pruned models. We summarize the key factors that influence pruning effectiveness as follows:

**Imbalanced Weight Magnitude Distribution.** LLMs' weight magnitudes distribution is imbalanced, with certain input or output channels exhibiting abnormally large or small magnitudes. This imbalance leads to sub-optimal pruning decisions, as weights within a channel are either predominantly preserved or pruned, regardless of their actual importance to the model's performance.

**Disproportionate Influence of Outliers.** A few activation outliers can disproportionately inflate a channel's norm,

[1]Advanced Micro Devices, Inc [2]Institute of Computing Technology, CAS, China [3]University of Chinese Academy of Sciences, Beijing, China [4]Zhongguancun Laboratory. Correspondence to: Ying Wang <wangying2009@ict.ac.cn>.

*Proceedings of the 42nd International Conference on Machine Learning*, Vancouver, Canada. PMLR 267, 2025. Copyright 2025 by the author(s).

leading to biased pruning. For instance, fewer than 1% of outliers can increase a channel's norm by over 5×, causing excessive pruning of outlier-free channels.

To address these limitations, we propose **BaWA**, a novel pruning mask selection method that systematically Balances the contributions of Weight and Activation distributions. BaWA introduces two key innovations. Firstly, to address the imbalanced weight magnitude distribution, BaWA normalizes the weight magnitudes across both input and output channels, contributing to a fairer pruning mask selection. Furthermore, to mitigate the disproportionate influence of outliers, BaWA introduces learnable power factors that control the impact of outliers on the pruning metric, ensuring that channels without outliers are not unevenly penalized. Moreover, BaWA employs a zeroth-order gradient-based optimization strategy to efficiently and automatically search for the optimal hyper-parameters of normalization and regularization, enabling the method to identify better pruning masks in just a few minutes. Notably, the reliable pruning metric provided by BaWA is orthogonal to conventional weight adjustment methods, and the combination of the two can bring better pruning outcomes. Through extensive experiments, we demonstrate that BaWA significantly outperforms existing state-of-the-art pruning methods across a variety of LLMs and language benchmarks. For instance, applying BaWA to induce 2:4 sparsity in Mistral-7B reduces language comprehension perplexity by 2.49 and improves average downstream task accuracy by 3.08% compared to the previous SOTA method Wanda.

In summary, the key contributions of this paper are:

- A comprehensive analysis of the limitations of current pruning metrics, highlighting the bias introduced by the symbolic combination of weight and activation.

- The introduction of BaWA, a novel pruning metric that balances weight and activation distributions through magnitude normalization and outlier regularization.

- An efficient optimization strategy for identifying optimal hyper-parameters for normalization and regularization, enabling BaWA to achieve superior pruning performance with minimal computational overhead.

- Experimental results demonstrate that BaWA successfully improves the performance of pruned LLMs compared to the state-of-the-art (SOTA) pruning methods.

## 2. Background & Motivation

### 2.1. LLM Pruning

The growing complexity of Transformer-based language models, which now scale to hundreds of billions of parameters, has intensified the demand for effective and efficient model pruning methods (Hassibi et al., 1993; Han et al., 2015b). These methods can be broadly categorized into structured and unstructured pruning.

Structured pruning (An et al., 2024; Xia et al.; Ma et al., 2023) removes entire substructures or weight groups—such as layers (Ling et al., 2024), FFN neurons (Ma et al., 2023), MHA heads, or embedding dimensions (Sreenivas et al., 2024)—enabling hardware-agnostic efficiency gains. However, due to its coarse-grained nature, structured pruning often leads to significant accuracy degradation, typically limiting the applicable sparsity ratio to 15%–30%. Post-pruning fine-tuning can help recover performance at higher sparsity levels.

In contrast, unstructured pruning (Frantar & Alistarh, 2023; Sun et al., 2023) eliminates individual weight elements and stores them in a compressed format. When combined with decompression techniques (for memory optimization) or hardware acceleration (e.g., 2:4 sparse tensor cores), unstructured sparsity can also deliver substantial efficiency improvements. Thanks to its fine-grained approach, unstructured pruning generally preserves model accuracy more effectively, allowing sparsity ratios exceeding 50%—or even stricter patterns like 2:4 sparsity. Further training techniques, such as PEFT (Lu et al., 2024) and STE (Ma et al., 2024), can also enhance pruned models.

Given these advantages, this paper focuses on optimizing unstructured pruning metrics.

### 2.2. Pruning Metric

The one-shot post-training pruning process primarily comprises two stages: pruning mask selection and weight reconstruction. Previous studies (Frantar & Alistarh, 2023; Li et al., 2023b; Liu et al., 2021) mainly focus on efficient weight reconstruction methodologies, while simply selecting the pruning mask based on weight magnitude. However, due to the presence of outliers, a larger magnitude does not necessarily indicate greater importance for a weight. As a result, directly pruning weights based solely on weight magnitude can result in the removal of important weights and lead to a significant performance drop for pruned LLMs. In fact, the effectiveness of pruning in large language models (LLMs) critically depends on the pruning metric (Li et al., 2023a; Frantar & Alistarh, 2023; Fang et al.), which quantifies the importance of each weight and guides the decision of which weights to remove. A well-designed pruning metric is essential for maintaining model performance after pruning, as it directly influences the quality of the pruned model.

As presented in Table 1, current pruning metrics utilize the symbolic combination of weight (Han et al., 2015a), activation (Sun et al., 2023; Zhang et al., 2024) and even

Table 1: The existing pruning metrics tailored for LLMs. **SC** denotes the symbolic combination operation, $G$ denotes the gradient, and $\sigma$ denotes the min-max scaling operation.

| Method | Pruning Metric $S$ | SC |
|---|---|---|
| Mangnitude | $\|W_{ij}\|$ | ✓ |
| Wanda | $\|W_{ij}\| \cdot \|\|X_j\|\|_2$ | ✓ |
| GBLM-Pruner | $\|W_{ij}\| \cdot \|\|G_j\|\|_2$ | ✓ |
| Pruner-Zero | $\|\|W_{ij}\| \times \|W_{ij}\|\| \times \sigma(\|G_j\|)$ | ✓ |
| BaWA | $(\frac{1}{\|\|W_j\|\|_2^{\theta_1}} + \frac{1}{\|\|W_i\|\|_2^{\theta_2}}) \cdot \|W_{ij}\| \cdot \|\|X_j\|\|_2^{\theta_3}$ | ✗ |

gradient (Dong et al.; Das et al., 2023) to determine the importance of weights. For example, Wanda defines the pruning metric as:

$$S_{ij} = |W_{ij}| \cdot ||X_j||_2, \qquad (1)$$

where $W \in \mathbb{R}^{C_{out} \times C_{in}}$ is the weight of a linear layer and $X \in \mathbb{R}^{B \cdot Q \times C_{in}}$ is the input activation with batch size $B$ and sequence length $Q$. $|\cdot|$ is the absolute value operator and $||X_j||_2$ represents the $\ell_2$ norm of $j$th features aggregated across $B \cdot Q$ different tokens. Wanda prunes those weights that have lower final scores $S$. GBLM-Pruner (Das et al., 2023) further utilizes gradient to adjust the pruning metric; however, the improved performance is limited. To improve the pruning performance, Pruner-Zero (Dong et al.) proposes an automatic framework for searching symbolic pruning metrics using genetic programming.

## 2.3. Motivation

However, existing pruning metrics rely on simplistic, symbolic combinations of weight, activation, and gradient values, failing to account for the intricate and heterogeneous distributions of weights and activations in LLMs. This oversight often leads to sub-optimal pruning decisions, resulting in significant performance degradation and inefficiencies in model compression.

To address this gap, we conduct a comprehensive analysis and visualization of weight and activation distributions in LLaMA-7B, as illustrated in Figure 1. Our investigation reveals two critical phenomena that challenge conventional assumptions and motivate the development of a more robust pruning methodology:

**Imbalanced Distribution of Weight Magnitude in LLM.** Recent studies on LLM compression often assume that weights follow a normal distribution, leading many to overlook the nuanced analysis of weight distributions (Xiao et al., 2023; Yin et al., 2023; Shao et al., 2023). However, as demonstrated in Figure 1a and 1b, weight magnitudes exhibit significant imbalances across channels. Specifically, certain input or output channels contain weights that are either abnormally large or small, creating a skewed distribution. This imbalance results in a scenario where most weights within a channel are either entirely preserved or pruned during compression. For unstructured and N:M sparsity, this concentration of pruning decisions on specific channels leads to sub-optimal sparsity patterns, as the pruning process fails to account for the heterogeneity in weight importance. This observation underscores the need for a pruning metric that explicitly considers the imbalanced nature of weight distributions, rather than relying on oversimplified assumptions.

**Disproportionate Impact of the Small Set of Outliers.** When constructing pruning metrics, existing methods such as Wanda (Equation 1) compute channel-wise norms (e.g., the $\ell_2$-norm) to evaluate the importance of weights. However, our analysis reveals that a small subset of outliers in activations can disproportionately influence these norm values. As shown in Figure 1c, even when outliers constitute less than $1\%$ of a channel's activations, they can inflate the channel's norm value by up to $5\times$ compared to the average. This disproportionate impact introduces significant bias into the pruning metric, as channels with outliers are erroneously prioritized for preservation, while others are undervalued. Consequently, pruning decisions based on such metrics become skewed, leading to sub-optimal sparsity patterns and degraded model performance.

It is necessary to mention that the two phenomena are widely present in various LLMs (qwe, 2024; Jiang et al., 2023; Touvron et al., 2023b). Consequently, we argue that a balanced pruning metric must consider both the weight magnitude distribution and the impact of outliers in activations. Such a metric should normalize the weight magnitudes across channels to mitigate the effects of imbalanced distributions and regulate the influence of outliers to ensure a fair evaluation of each weight's importance. By balancing these factors, a more accurate and effective pruning metric can be developed, enabling better pruning decisions and improved model performance.

## 3. Methods

### 3.1. Magnitude Normalization

According to our observation in Section 2.3, the weight values of different channels (including both input and output channels) are quite different. Consequently, when establishing the pruning metric, it is imperative to consider not only the per-channel norm values of activation but also the norm values within the weight. This comprehensive consideration ensures a more balanced approach to determining the pruning mask, which is critical for maintaining the pruning performance. We present the details of the input and output channel normalization as follows.

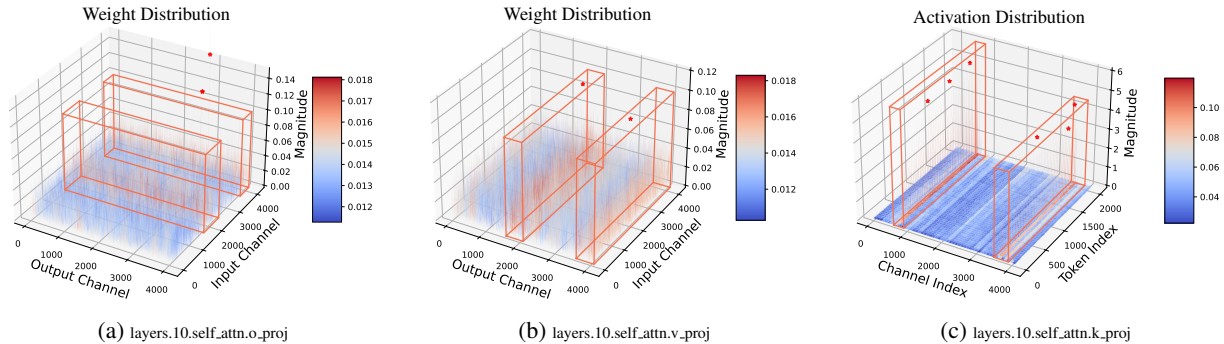

Figure 1: The visualization of the weight and activation magnitude distribution in the 10th layer of LLaMA-7B. The z-axis of the 3-D figure represents the magnitude of the weights and activations. Observations: (1) The distributions vary greatly not only across the channel dimension of activations but also across the input and output channel dimensions of weights. The three figures below demonstrate that the norms of different channels vary significantly. (2) The norm of one channel is often heavily influenced by outliers, despite their small proportion. Figure (c) is particularly notable, demonstrating that several outliers lead to a significant norm. ★ denotes outliers.

**Input Channel Normalization.** In contrast to the activation values, the magnitude of weight exhibits a relatively more gradual variation. Consequently, as presented in Equation (1), compared to the weight magnitude $|\boldsymbol{W}_{ij}|$, the proposed pruning metric in Wanda mainly focuses on capturing the fluctuations in activation values $||\boldsymbol{X}_j||_2$. This strategy emphasizes that the dynamic range of activation is more susceptible to outliers, which can significantly influence the pruning performance. However, our observations indicate that the per-channel norm value within the weights also exhibits considerable variation. Therefore, we argue that the pruning metric should not be exclusively skewed towards outliers in activation. It is equally important to take into account the magnitude of weights themselves. Accordingly, we introduce the $\ell_2$-norm weight value for each input channel to mitigate the excessive focus on activation values. We incorporate input channel normalization and redefine the pruning metric as outlined below.

$$\boldsymbol{S}_{ij}^{(ICN)} = |\boldsymbol{W}_{ij}| \cdot \frac{1}{||\boldsymbol{W}_j||_2} \cdot ||\boldsymbol{X}_j||_2. \qquad (2)$$

**Output Channel Normalization.** Recent studies have revealed that outliers tend to occur in specific channels of activation (Yuan et al., 2023; Hooper et al., 2024; Yao et al., 2022). We notice that the presence of outliers in specific channels is closely related to the magnitude of weights across different output channels. Therefore, for these channels, it is critical to pay attention to the inherent magnitude of the original weights. Building on these insights, we propose output channel normalization, a technique that leverages the $\ell_2$-norm of weight values for each output channel to alleviate the undue emphasis on activation values. Especially when the $\ell_2$-norm of an output channel is substantially large, our proposed pruning metric shifts its focus more towards the magnitude of the weight values themselves. The pruning metric with output channel normalization is as follows:

$$\boldsymbol{S}_{ij}^{(OCN)} = \frac{1}{||\boldsymbol{W}_i||_2} \cdot |\boldsymbol{W}_{ij}| \cdot ||\boldsymbol{X}_j||_2. \qquad (3)$$

It is important to note that during unstructured and N:M pruning, we often prune each output channel by the same sparsity ratio, rendering the consideration of output channel normalization alone unproductive. Therefore, we need to jointly consider input normalization and output channel normalization to form the final magnitude normalization metric $\boldsymbol{S}_{ij}^{(MN)} = \boldsymbol{S}_{ij}^{(ICN)} + \boldsymbol{S}_{ij}^{(OCN)}$. As illustrated in Figure 2a, adopting magnitude normalization helps to make the distribution of pruning mask more balanced on different channels, improving the performance of pruned LLMs.

### 3.2. Outlier Regularization

Our analysis in Section 2.3 has revealed that a small set of outliers can significantly influence the norm value of a channel. According to our evaluation, when using Wanda as the pruning metric for LLaMA-7B (Touvron et al., 2023a), a considerable number of channels were eliminated across various layers (even exceeding 10% in specific layers). That is to say, when adopting the typical norm value of each channel as a part of the pruning metric, channels devoid of outliers are susceptible to being inadvertently compromised during the pruning process (Zhang et al., 2024), as illustrated in the left part of Figure 2b.

The outlier values have been unduly emphasized in the pruning process (Wei et al., 2023). It is necessary to reduce the proportion of weights pruned from channels that do not contain outliers. To address this, we propose a novel

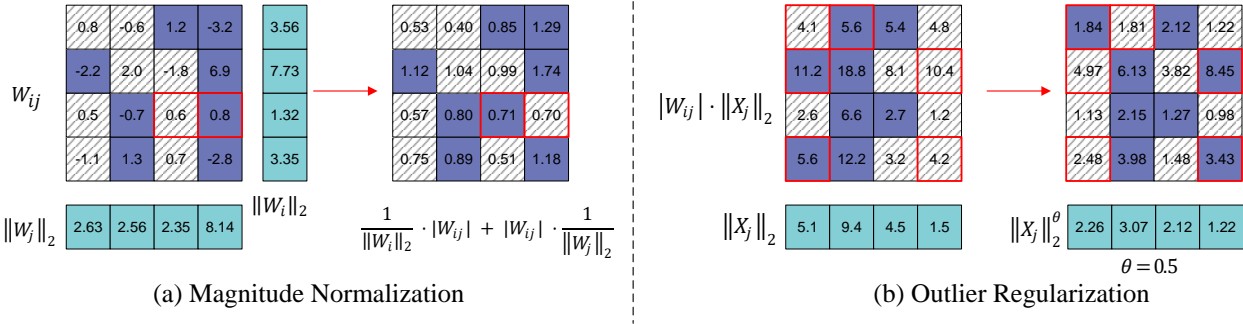

(a) Magnitude Normalization          (b) Outlier Regularization

Figure 2: Illustration of our proposed magnitude normalization and outlier regularization methods. The leftmost figure represents the naïve pruning approach, which prunes weights with smaller magnitudes to satisfy the 2:4 constraints. The changed pruned positions are highlighted with red boxes. (a) magnitude normalization re-evaluates the importance of weight magnitude by considering the effect of norm values across input and output channel dimensions. (b) Outlier regularization suppresses the effect of larger activation norms and leads to different weight pruning masks.

strategy, outlier regularization, to reduce the impact of outliers. Outlier regularization aims to diminish the variability in magnitude across different channels, thereby ensuring a more equitable and balanced pruning strategy. Specifically, when defining the pruning metric, we introduce a novel pruning parameter $\theta$, to serve as the power factor for the norm value. Specifically, the pruning metric with outlier regularization can be modified as:

$$\mathbf{S}_{ij}^{(OR)} = |\boldsymbol{W}|_{ij} \cdot ||\boldsymbol{X}_j||_2^{\theta}. \tag{4}$$

The proportion of unreasonable norm values generated by outliers is not solely present in activation. As revealed in previous work (Nrusimha et al., 2024), outliers also exist in the distribution of weight values. Therefore, we apply outlier regularization not only to the norm of activation but also to the per-channel norm values of the weights. Incorporating the two methods previously discussed, we can formulate the final pruning metric, BaWA, as follows:

$$\boldsymbol{S}_{ij} = (\underbrace{|\boldsymbol{W}_{ij}| \cdot \frac{1}{||\boldsymbol{W}_j||_2^{\theta_1}}}_{\text{input channel normalization}} + \underbrace{\frac{1}{||\boldsymbol{W}_i||_2^{\theta_2}} \cdot |\boldsymbol{W}_{ij}|}_{\text{output channel normalization}}) \cdot ||\boldsymbol{X}_j||_2^{\theta_3},$$

$$\tag{5}$$

where $\theta_1$, $\theta_2$, and $\theta_3$ represent the power factors for the input channel, output channel, and activation norm, respectively.

### 3.3. Efficient Power Factor Search

Based on the proposed pruning metric in Equation (5), each linear layer has three power factors, $\theta_1$, $\theta_2$, and $\theta_3$, to be determined. As different layers often exhibit various distributions, the optimal values for the three factors may vary accordingly. As the number of layers increases, the search space expands exponentially, resulting in an inefficient search process. We introduce the design of an efficient search method from two perspectives: search target and

parameter optimization algorithm. According to our evaluation, the proposed efficient search process can be completed in about 16 minutes for LLaMA2-70B within a single GPU.

**Search Target.** A straightforward way to prevent the search space from growing exponentially is to replace the global search target with a local search target. Previous works (Frantar & Alistarh, 2023; Nagel et al., 2020; Frantar & Alistarh, 2022; Hassibi et al., 1993) optimize the pruning loss layer-by-layer, reducing the search space from $C^L$ to $C \cdot L$. $C$ represents the search complexity for one layer and $L$ is the number of linear layers in one model. However, this approach may lack consideration of the interaction between layers. Other works (Shao et al., 2023; Li et al., 2020) employ the idea of block-wise optimization to incorporate interaction between layers within a single block. Each transformer block is structurally identical and relatively independent, making it well-suited for holistic optimization. Suppose a Transformer block contains $m$ linear layer, the search space is $C^m \cdot L/m$. We formulate the search target as follows:

$$\Theta^* = \underset{\Theta}{\arg\min} \, \mathcal{L}(\Theta; \boldsymbol{X}), \tag{6}$$

$$\mathcal{L}(\Theta; \boldsymbol{X}) = \|\mathrm{RMSNorm}\,(\mathcal{F}(\mathbb{W}; \boldsymbol{X}))$$
$$- \mathrm{RMSNorm}\,(\mathcal{F}(\mathbb{W} \odot \mathbb{M}; \boldsymbol{X}))\|_2^2, \tag{7}$$

$$\mathbb{M} = \mathbb{S} > \mathrm{top_k}(\mathbb{S}), \tag{8}$$

where $\mathcal{F}$ represents the computation of a single transformer block, which contains a self-attention and a feed-forward network. $\mathbb{W}$ is the weights for $m$ linear layers and $\boldsymbol{X}$ is the input activation of the block. $\Theta$ consists of $m$ sets of $(\theta_1, \theta_2, \theta_3)$. The pruning mask $\mathbb{M}$ is determined by our pruning metric $\mathbb{S}$ in Equation (5), where weights with smaller pruning metric are pruned by $\mathrm{top_k}$ function. The symbol

$\odot$ represents the element-wise multiplication, and $|| \cdot ||_2$ is the $\ell_2$ norm function. Different blocks may exhibit varying sensitivities to pruning, resulting in significant fluctuations in pruning error across blocks. To address this issue, we adopt the RMSNorm (Touvron et al., 2023a) after the output feature $\boldsymbol{O}$, containing $n$ elements, to balance the error disparities among different transformer blocks:

$$
\begin{aligned}
\text{RMSNorm}(\boldsymbol{O}) &= \frac{o_i}{\text{RMS}(\boldsymbol{O})}, \\
\text{where} \quad \text{RMS}(\boldsymbol{O}) &= \sqrt{\frac{1}{n} \sum_{i=1}^{n} o_i^2}.
\end{aligned}
\tag{9}
$$

We demonstrate the effectiveness of this error normalization strategy in Appendix 4.5.

**Parameter Optimization Algorithm.** The reliance on empirical or grid search strategies (Zhang et al., 2024; Lin et al., 2023), while common, can become inefficient as the search space expands. Gradient-based optimization is another typical strategy for parameter tuning, especially in the studies of neural network compression (Li et al., 2020; Nagel et al., 2020; Shao et al., 2023). Unfortunately, obtaining $\mathbb{M}$ involves a non-differentiable $\text{top}_k$ function, which poses an obstacle to computing the gradient of power factors.

Recent studies utilize a zeroth-order optimizer to enable non-differentiable parameter tuning, while also conserving GPU memory and improving efficiency (Malladi et al., 2024; Spall, 1992). Motivated by this research, we use a zeroth-order optimizer to estimate the corresponding gradient for each power factor. Specifically, the gradients of $\Theta$ can be described as:

$$
\begin{aligned}
\hat{\nabla}\mathcal{L}(\Theta; \boldsymbol{X}) &= \frac{\mathcal{L}(\Theta + \epsilon z; \boldsymbol{X}) - \mathcal{L}(\Theta - \epsilon z; \boldsymbol{X})}{2\epsilon} z \\
&\approx z z^\top \nabla\mathcal{L}(\Theta; \boldsymbol{X}).
\end{aligned}
\tag{10}
$$

Here, $\mathcal{L}$ denotes the error loss in Equation 6, $z$ is a sampled data that satisfies $z \sim \mathcal{N}(0, I_d)$, and $\epsilon$ is the perturbation scale. $I_d$ is an identity matrix with the dimension $d = 3m$, as each layer has three power factors to be determined. Following setting in MeZO (Malladi et al., 2024), $\epsilon$ in our experiments is set as $0.01$. We perturb the power factors at each step and obtain the corresponding gradients. We then update them based on these gradients. It is often necessary to repeat this iteration process times to efficiently converge to the optimal solution. The details on updating these parameters are shown in Algorithm 2 of Appendix E.

# 4. Experiments

## 4.1. Experimental Setup

**Power Factor Search.** Due to the presence of numerous outliers in the activation distribution and only a minimal number of outlier values in the weights, we empirically initialize the power factors for the weight norm as $1$, while $0.5$ for the activation norm. To optimize these pruning parameters, we utilize a typical SGD optimizer without weight decay. The learning rate is set as $0.2$. Following the setting in Omniquant (Shao et al., 2023), we employ a calibration dataset consisting of 128 randomly selected 2048-token segments from C4 (Raffel et al., 2020). We set the batch size as 16 with only 2 epochs for the optimization process. We further explore the different optimization settings in our evaluation. The whole optimization process can be completed in about 16 minutes for LLaMA2-70B (Touvron et al., 2023b) within a single GPU, which is less than that of SparseGPT.

**Models and Evaluation.** We evaluate BaWA on widely adopted LLMs, including LLaMA (7B-65B) (Touvron et al., 2023a), LLaMA2 (7B-70B), Mistral-7B (Jiang et al., 2023) and Qwen2-72B (qwe, 2024). We measure the performance of pruned models on seven zero-shot tasks. We further evaluate the five-shot performance of MMLU in the Appendix. For zero-shot evaluation, we use seven tasks from EleutherAI LM Harness (Gao et al., 2021), including HellaSwag (Clark et al., 2018), PIQA (Bisk et al., 2020), ARC (Clark et al., 2018), BoolQ (Clark et al., 2019), RTE and Winogrande. Following previous works on LLM compression (Frantar & Alistarh, 2023; Xiao et al., 2023), we also evaluate the perplexity on the WikiText-2 (Merity et al., 2016).

**Baselines.** We compare BaWA with various methods including (1) Magnitude-based pruning (Han et al., 2015b) that discards weights based on their magnitudes. (2) SparseGPT (Frantar & Alistarh, 2023) that utilizes second-order Hessian inverses to ascertain unimportant weights. (3) Wanda (Sun et al., 2023) that removes weights with the smallest magnitudes multiplied by the corresponding input activation norms. Moreover, we also compare BaWA with diverse existing pruning metrics, as depicted in Table 1, that use the symbolic combination of weight, activation and gradients to demonstrate the necessity of a balanced distribution of weight and activation.

## 4.2. Language Modeling

### 4.2.1. BAWA ON VARYING SPARSITY.

We investigate the efficacy of BaWA when pruning LLMs with varying pruning rates. Table 2 shows that BaWA performs better than Wanda at different sparsity levels. Particularly, this improvement becomes increasingly evident as the

Table 2: WikiText-2 perplexity performance of BaWA and Wanda for different LLMs at varying sparsity rates.

| | LLaMA-7B | | | LLaMA-13B | | | LLaMA2-70B | | | Qwen2-72B | | |
|---|---|---|---|---|---|---|---|---|---|---|---|---|
| Sparsity | 60% | 70% | 80% | 60% | 70% | 80% | 60% | 70% | 80% | 60% | 70% | 80% |
| Wanda | 10.57 | 74.79 | 4.80e3 | 8.69 | 51.94 | 4.95e3 | 4.97 | 10.23 | 149.76 | 6.26 | 9.00 | 40.50 |
| BaWA | **10.00** | **57.84** | **3.95e3** | **7.67** | **33.83** | **4.10e3** | **4.56** | **8.71** | **125.71** | **6.03** | **8.17** | **31.89** |

sparsity level grows. For instance, we can achieve a reduction of **16.95** and **18.11** for LLaMA1-7B and 13B models under 70% sparsity. For large models such as LLaMA2-70B and Qwen2-72B, BaWA can still achieve a reduction of **24.05** and **8.61** under 80% sparsity compared to Wanda.

Table 3: WikiText-2 perplexity of various pruned LLMs for N:M sparsity. We compare BaWA with existing prune-only methods, including Magnitude (Han et al., 2015a), Wanda (Sun et al., 2023), GBLM (Das et al., 2023), RIA (Zhang et al., 2024) and Pruner-Zero (Dong et al.), as well as weight reconstruction methods, including SparseGPT (Frantar & Alistarh, 2023), DSnoT (Zhang et al., 2023) and ADMM-Iter (Boža, 2024). We further combine BaWA with the SOTA weight reconstruction method ADMM-Iter, denoted as BaWA+ADMM. In our evaluation, **bold** denotes the best performance and underline denotes the second best performance.

| | | LLaMA2 | | Mistral | Qwen2 |
|---|---|---|---|---|---|
| Method | Sparsity | 13B | 70B | 7B | 72B |
| Dense | 0% | 4.57 | 3.12 | 5.25 | 4.94 |
| Magnitude | 4:8 | 6.76 | 5.54 | 9.21 | 8.14 |
| SparseGPT | 4:8 | 6.60 | 4.59 | 8.07 | 5.97 |
| Wanda | 4:8 | 6.55 | 4.47 | 8.41 | 5.86 |
| GBLM | 4:8 | 6.54 | 4.49 | 8.31 | 5.85 |
| RIA | 4:8 | 6.29 | 4.37 | 8.27 | 5.81 |
| Pruner-Zero | 4:8 | 6.75 | 4.45 | 8.11 | 5.85 |
| DSnoT | 4:8 | 6.43 | 4.41 | 7.93 | 5.79 |
| ADMM-Iter | 4:8 | 6.37 | 4.35 | 7.79 | 5.77 |
| BaWA | 4:8 | 6.16 | 4.32 | 7.54 | 5.74 |
| BaWA + ADMM | 4:8 | **6.07** | **4.24** | **7.36** | **5.65** |
| Magnitude | 2:4 | 8.33 | 6.33 | 13.57 | 8.15 |
| SparseGPT | 2:4 | 8.32 | 5.40 | 10.52 | 6.51 |
| Wanda | 2:4 | 8.27 | 5.16 | 12.37 | 6.31 |
| GBLM | 2:4 | 8.80 | 5.47 | 10.97 | 6.39 |
| RIA | 2:4 | 7.77 | 5.11 | 10.51 | 6.28 |
| Pruner-Zero | 2:4 | 7.41 | 4.81 | 10.49 | 6.33 |
| DSnoT | 2:4 | 8.09 | 5.11 | 10.24 | 6.26 |
| ADMM-Iter | 2:4 | 7.78 | 5.19 | 10.29 | 6.21 |
| BaWA | 2:4 | 7.13 | 4.84 | 9.88 | 6.14 |
| BaWA + ADMM | 2:4 | **7.04** | **4.71** | **9.53** | **6.01** |

### 4.2.2. PERPLEXITY WITH N:M SPARSITY.

As presented in Table 3, we compare the perplexity of various pruned LLMs with existing LLM pruning methods. As one can notice, BaWA consistently achieves better performance when compared with the other strong baselines without introducing weight reconstruction. Specifically, employing BaWA as the pruning metric achieves up to a 1.23 reduction in perplexity in the 2:4 sparsity (LLaMA2-13B) relative to Wanda. This result suggests that exact and effective sparse sub-networks exist for LLMs, and finds the appropriate metric is non-trivial but important for LLM pruning. Furthermore, BaWA can also combine with existing weight reconstruction pruning methods. As our experimental results depict, BaWA + ADMM can achieve the best performance. We also present more evaluations on diverse LLMs and sparse patterns in Appendix B.

### 4.3. Zero-Shot Tasks

We further compare BaWA with baselines on 7 typical zero-shot tasks. As shown in Table 10, we present the mean zero-shot accuracy of the pruned models within the LLaMA, LLaMA2 families, Mistral-7B and Qwen2-72B (the detailed performance of each task can be found in Table 11 and Table 12). We implement the lm-eval-harness (Gao et al., 2021) for all zero-shot tasks, with the report including both the accuracy results on each benchmark and the overall average accuracy. As it illustrates, BaWA consistently outperforms Wanda. For LLaMA2-70B, the pruned model under 50% sparsity achieves better accuracy than the original dense model. In certain cases, SparseGPT achieves higher accuracy than our proposed BaWA, suggesting that weight reconstruction can further enhance the performance of pruned LLMs. Fortunately, our proposed pruning metric method can seamlessly integrate with novel weight reconstruction techniques (Zhang et al., 2023; Kwon et al., 2022), which will be detailed in Appendix F.

### 4.4. Ablation Study

In this paper, we introduce multiple strategies to achieve a balanced orchestration of weight and activation. To verify the efficacy of each strategy, we conducted a comprehensive ablation study, as shown in Table 5. We evaluate the impact of magnitude normalization and outlier regularization

Table 4: Mean zero-shot accuracies (%) of pruned LLaMA, LLaMA2, Mistral-7B and Qwen2-72B models on 7 zero-shot tasks. BaWA performs competitively against the prior best methods SparseGPT and Wanda.

| Method | Weight Update | Sparsity | LLaMA | | | | LLaMA2 | | | Mistral-7B | Qwen2-72B |
| | | | 7B | 13B | 30B | 65B | 7B | 13B | 70B | | |
|---|---|---|---|---|---|---|---|---|---|---|---|
| Dense | - | 0% | 59.99 | 62.59 | 65.38 | 66.97 | 59.71 | 63.03 | 67.08 | 64.30 | 69.82 |
| Magnitude | ✗ | 50% | 46.94 | 47.61 | 53.83 | 62.74 | 51.14 | 52.77 | 60.93 | 55.87 | 60.66 |
| SparseGPT | ✓ | 50% | 54.94 | 58.61 | 63.09 | 66.30 | 56.24 | 60.57 | 67.28 | 59.34 | 68.11 |
| Wanda | ✗ | 50% | 55.13 | 59.33 | 63.60 | 66.67 | 56.24 | 60.04 | 67.03 | 58.93 | 66.41 |
| BaWA | ✗ | 50% | **55.27** | **59.97** | **64.12** | **67.21** | **57.02** | **60.67** | **67.81** | **60.17** | **69.11** |

Table 5: Ablation study on our proposed methods. We propose several strategies to improve the pruning mask metric for LLMs. We evaluate the performance improvements provided by these strategies, respectively. Only adopting Output Channel Normalization does not alter the pruning position, so we don't present its experimental results.

| Method | LLaMA2 & Qwen2 (50%) | | | LLaMA2 & Qwen2 (4 : 8) | | | LLaMA2 & Qwen2 (2 : 4) | | |
| | 13B | 70B | 72B | 13B | 70B | 72B | 13B | 70B | 72B |
|---|---|---|---|---|---|---|---|---|---|
| Wanda | 5.56 | 3.98 | 5.48 | 6.55 | 4.47 | 5.86 | 8.27 | 5.16 | 6.31 |
| Input Channel Normalization | 5.47 | 3.89 | 5.48 | 6.38 | 4.42 | 5.84 | 7.93 | 5.13 | 6.30 |
| Magnitude Normalization | 5.45 | 3.88 | 5.44 | 6.27 | 4.41 | 5.81 | 7.74 | 5.04 | 6.27 |
| Outlier Regularization (0.5) | 5.46 | 3.90 | 5.46 | 6.20 | 4.39 | 5.77 | 7.54 | 4.95 | 6.21 |
| BaWA w/o Automatic Search | 5.45 | 3.88 | 5.43 | 6.27 | 4.41 | 5.80 | 7.74 | 5.05 | 6.23 |
| BaWA w/ Automatic Search | **5.42** | **3.84** | **5.41** | **6.16** | **4.32** | **5.74** | **7.13** | **4.84** | **6.14** |

on LLaMA2-13B, LLaMA2-70B and Qwen2-72B, respectively. The power factor is set as 0.5 for the outlier regularization. Compared to Wanda, both magnitude normalization and outlier regularization reduce perplexity to some extent. Specifically, magnitude normalization has a greater impact on performance improvement in unstructured pruning, while outlier regularization has a more significant effect on perplexity under N:M sparsity conditions. It is noteworthy that even without careful search, our BaWA method can easily outperform Wanda. Furthermore, automating the search for the power factors $\Theta$ consistently reduces model perplexity. As presented in Appendix C, different transformer blocks, different linear layers, and different sparse patterns prefer quite different power factors. It is necessary to find the appropriate pruning parameters for every linear layer.

**Power Factor Analysis.** We further analyze how different power factor settings affect the pruning performance. As shown in Table 6, different power factor settings exhibit significant sensitivity ($\pm 24.1\%$ PPL variance). Additionally, the optimized power factor setting effectively reduces perplexity by 12.3% compared to the best-fixed scale ($\theta = 0.5$).

### 4.5. Effectiveness Analysis

**Roubustness Analysis.** We analyze the impact of sample size on the performance of the pruned model for LLaMA2-13B. As depicted in Figure 3, it is evident that BaWA is almost insensitive to the number of samples. Even with

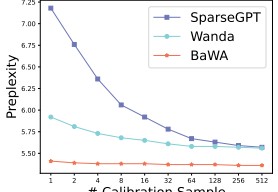
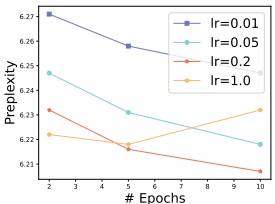

Figure 3: The robustness analysis of pruning methods on calibration samples.

Figure 4: Effect analysis on learning rate (lr) and epochs for parameter optimization.

only one sample, satisfactory results could be achieved. The remarkable robustness exhibited by BaWA primarily stems from the outlier regularization strategy we proposed, which effectively suppresses the impact of anomalies in the sampling process on the pruning results. We further analyze the performance variance when adopting different random seeds, as illustrated in Appendix D.

**Hyper-Parameter Exploration** We also compare results under different hyper-parameter conditions to investigate the impact of varying hyper-parameters (learning rate and epochs) on the search outcomes during the parameter optimization process. As shown in Figure 4, we notice that when the learning rate is small ($\leq 0.2$), perplexity decreases as the number of epochs increases. However, a minimal learning rate leads to slow convergence, significantly im-

Table 6: Power Factor Analysis.

| Scaling Strategy | $\theta_1$ (Input) | $\theta_2$ (Output) | $\theta_b$ (Activation) | PPL | $\Delta$ vs. Best Fixed (Fixed at 0.5) |
|---|---|---|---|---|---|
| Fixed Scales $\theta = 0.1$ | 0.1 | 0.1 | 0.1 | 8.92 | +24.1% |
| Fixed Scales $\theta = 0.5$ | 0.5 | 0.5 | 0.5 | 7.18 | **+0% (baseline)** |
| Fixed Scales $\theta = 1.0$ | 1.0 | 1.0 | 1.0 | 7.53 | +4.9% |
| BaWA Optimized | 0.42 | 0.51 | 0.38 | **6.30** | **-12.3%** |

proving the search cost. Conversely, a larger learning rate makes maintaining a sustained model perplexity reduction with increased epochs difficult. Therefore, in this paper, we set the learning rate at 0.2 and use only epoch = 2 for parameter search. While larger epochs could further enhance the performance of the pruned LLM, the additional search cost makes such an attempt unnecessary.

Table 7: Time overhead with BaWA for pruning LLaMA-2 model family. The performance is evaluated on a single NVIDIA A100 40GB GPU.

| Method | 7B | 13B | 70B |
|---|---|---|---|
| Wanda | 0.4 s | 0.8s | 2.2 s |
| SparseGPT | 3.1 min | 5.2 min | 22.3 min |
| BaWA w/o Search | 0.5 s | 0.8 s | 2.4 s |
| BaWA w/ Search | 1.9 min | 3.7 min | 16.2 min |

**Pruning Overhead.** We further analyze the computing cost of BaWA. Following Wanda (Sun et al., 2023), we report the time of total parameter optimization and pruning. Table 7 shows the quantitative wall-clock overhead evaluated on a single NVIDIA A100 GPU. We have evaluated the optimization overhead of BaWA on different models, both with and without parameter search. As shown in Table 7, one can obtain a high-performance pruned LLM within seconds. As depicted in Section 4.4, BaWA can easily outperform SOTA methods even without automatic optimization. Certainly, searching for suitable pruning parameters requires several minutes, which costs less optimization time than SparseGPT. For instance, approximately 16 minutes are needed on a single A100 GPU to search for appropriate parameters for the LLaMA-2-70B model. Therefore, users can adaptively adjust the strategy for parameter search based on their computational resources with limited searching costs.

**Sensitivity Analysis on Loss Function.** Figure 5 compares the loss across different layers for Wanda and BaWA on LLaMA2-13B. The loss value is evaluated according to Equation (6). It reveals that the initial layers exhibit smaller losses, indicating lower sensitivity and a reduced need for meticulous selection of pruning metrics. Conversely, the later layers demonstrate significant loss differentiation based on the choice of pruning strategies. Similarly, this suggests that using lower learning rates for parameter search in the

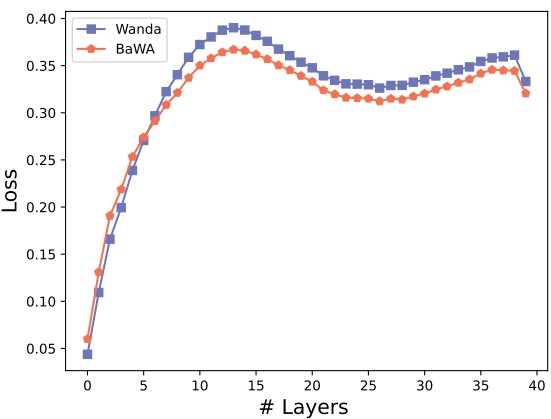

Figure 5: The loss value for different layers

initial layers can further reduce the loss in BaWA. Therefore, by finely adjusting the hyper-parameters during the search process, we hope to achieve better performance than the results reported in this paper.

## 5. Conclusion

In this work, we propose an improved pruning mask selection strategy for pruning LLMs. Our insight is that existing works ignore the imbalanced distribution in LLMs. To this end, we propose a novel algorithm, BaWA, which introduces magnitude normalization and outlier regularization to alleviate the impact of imbalanced distributions in both weight and activation. By introducing several learnable pruning parameters, BaWA typically discovers more effective pruning masks and enhances the existing pruning method. As the introduced parameters are non-differentiable, we carefully employ a forward-forward optimization strategy to find the solution efficiently. It is worth noting that the optimization process is highly efficient, requiring only 2 minutes for LLaMA2-7B and 16 minutes for LLaMA2-70B, for example. Experimental results demonstrate that BaWA is efficient and superior to existing methods across various language benchmarks under the same sparse ratio.

## Acknowledgments

We sincerely thank the anonymous reviewers for their insightful suggestions. Furthermore, we thank Lu Tian and Emad Barsoum from AMD for valuable discussions on the proposed metric design. This work was partially supported by the National Key R&D Program of China (Grant No. 2023YFB4404400) and the National Natural Science Foundation of China (Grant No. 62222411). Ying Wang is the corresponding author (wangying2009@ict.ac.cn).

## Impact Statement

This paper presents work whose goal is to advance the field of Machine Learning. There are many potential societal consequences of our work, none of which we feel must be specifically highlighted here.

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

## A. Hardware Efficiency

Table 8: GEMM Speedup with 2:4 sparsity for one transformer block of LLaMA-65B. The performance is evaluated on a single NVIDIA A100 40GB GPU.

| Layer | Dense GEMM | Sparse GEMM | Speedup |
|---|---|---|---|
| Q/K/V | 0.270 ms | 0.163 ms | **1.66**× |
| Att_Out | 0.097 ms | 0.072 ms | **1.35**× |
| Up/Gate | 0.487 ms | 0.285 ms | **1.71**× |
| Down | 0.247 ms | 0.176 ms | **1.40**× |

To illustrate the enhanced efficiency of pruned LLMs, we present the actual speedup on different operators for LLaMA-65B on a single A100 40GB GPU. Due to the lack of support for the 4:8 sparsity pattern in NVIDIA Ampere architecture, we only measure the latency of GEMM with 2:4 sparsity. As shown in Table 8, the model pruned by BaWA achieves $1.58\times$ speedup over dense FP16 GEMM baseline for the token generation stage. It is noteworthy that pruned LLMs can achieve acceleration not only on GPUs but also on modern CPUs such as AMD Ryzen 7 PRO 5850U.

## B. Detailed Results

Table 9: WikiText perplexity of various pruned LLMs. BaWA performs competitively against prior best methods SparseGPT and Wanda, specifically for N:M sparsity.

| Method | Weight Update | Sparsity | LLaMA | | | | LLaMA2 | | | Mistral-7B | Qwen2-72B |
|---|---|---|---|---|---|---|---|---|---|---|---|
| | | | 7B | 13B | 30B | 65B | 7B | 13B | 70B | | |
| Dense | - | 0% | 5.68 | 5.09 | 4.77 | 3.56 | 5.12 | 4.57 | 3.12 | 5.25 | 4.94 |
| Magnitude | ✗ | 50% | 17.29 | 20.21 | 7.54 | 5.90 | 14.89 | 6.37 | 4.98 | 7.87 | 7.84 |
| SparseGPT | ✓ | 50% | 7.22 | 6.21 | 5.31 | 4.57 | 6.51 | 5.63 | 3.98 | 6.46 | 5.51 |
| Wanda | ✗ | 50% | 7.26 | 6.15 | 5.24 | 4.57 | 6.42 | 5.56 | 3.98 | 6.54 | 5.48 |
| BaWA | ✗ | 50% | **7.08** | **6.01** | **5.06** | **4.40** | **6.23** | **5.37** | **3.84** | **6.39** | **5.41** |
| Magnitude | ✗ | 4:8 | 16.84 | 13.84 | 7.62 | 6.36 | 16.48 | 6.76 | 5.54 | 9.21 | 8.14 |
| SparseGPT | ✓ | 4:8 | 8.61 | 7.40 | 6.17 | 5.38 | 8.12 | 6.60 | 4.59 | 8.07 | 5.97 |
| Wanda | ✗ | 4:8 | 8.57 | 7.40 | 5.97 | 5.30 | 7.97 | 6.55 | 4.47 | 8.41 | 5.86 |
| BaWA | ✗ | 4:8 | **8.08** | **6.89** | **5.62** | **4.94** | **7.36** | **6.16** | **4.32** | **7.54** | **5.74** |
| Magnitude | ✗ | 2:4 | 42.13 | 18.37 | 9.10 | 7.11 | 54.59 | 8.33 | 6.33 | 13.57 | 8.15 |
| SparseGPT | ✓ | 2:4 | 11.00 | 9.11 | 7.16 | 6.28 | 10.17 | 8.32 | 5.40 | 10.52 | 6.51 |
| Wanda | ✗ | 2:4 | 11.53 | 9.58 | 6.90 | 6.25 | 11.02 | 8.27 | 5.16 | 12.37 | 6.31 |
| BaWA | ✗ | 2:4 | **10.32** | **7.94** | **6.37** | **5.61** | **9.93** | **7.13** | **4.84** | **9.88** | **6.14** |

**Results on WikiText2.** Table 9 shows the WikiText2 perplexity on various LLMs with different sparsity configurations, including 50% sparsity, 2:4 and 4:8 sparse patterns. As one can notice, BaWA always achieves the best performance on diverse LLMs and sparse patterns.

**Results on Zero-Shot Tasks.** For zero-shot results in Section 4.3, the seven evaluated zero-shot tasks are: BoolQ (Clark et al., 2019), RTE (Wang et al., 2018), HellaSwag (Zellers et al., 2019), WinoGrande (Sakaguchi et al., 2021), ARC Easy and Challenge (Clark et al., 2018), and OpenbookQA (Mihaylov et al., 2018). In this section, we also present the mean zero-shot accuracies of pruned LLMs under N:M sparsity, as shown in Table 10. To offer a more nuanced demonstration of the advantages of BaWA across various datasets, we present the performance evaluations for each zero-shot task of LLaMA2-13B and Mistral-7B, as shown in Table 11 and 12. Compared to Wanda and SparseGPT, BaWA demonstrates more consistent performance across various tasks, indicating its robustness. Additionally, BaWA shows high accuracy in the ARC tests, suggesting its strong adaptability to the task. Overall, utilizing BaWA can effectively enhance the precision of pruned LLMs in zero-shot tasks.

**Results on MMLU Validation.** To further validate the adaptability of BaWA across diverse datasets, we conduct additional

Table 10: Mean zero-shot accuracies (%) of pruned LLaMA, LLaMA2, Mistral-7B and Qwen2-72B models on 7 zero-shot tasks. BaWA performs competitively against the prior best methods SparseGPT and Wanda.

| Method | Weight Update | Sparsity | LLaMA | | | | LLaMA2 | | | Mistral-7B | Qwen2-72B |
|---|---|---|---|---|---|---|---|---|---|---|---|
| | | | 7B | 13B | 30B | 65B | 7B | 13B | 70B | | |
| Dense | - | 0% | 59.99 | 62.59 | 65.38 | 66.97 | 59.71 | 63.03 | 67.08 | 64.30 | 69.82 |
| Magnitude | ✗ | 50% | 46.94 | 47.61 | 53.83 | 62.74 | 51.14 | 52.77 | 60.93 | 55.87 | 60.66 |
| SparseGPT | ✓ | 50% | 54.94 | 58.61 | 63.09 | 66.30 | 56.24 | 60.57 | 67.28 | 59.34 | 68.11 |
| Wanda | ✗ | 50% | 55.13 | 59.33 | 63.60 | 66.67 | 56.24 | 60.04 | 67.03 | 58.93 | 66.41 |
| BaWA | ✗ | 50% | **55.27** | **59.97** | **64.12** | **67.21** | **57.02** | **60.67** | **67.81** | **60.17** | **69.11** |
| Magnitude | ✗ | 4:8 | 46.03 | 50.53 | 53.53 | 62.17 | 50.64 | 52.89 | 60.28 | 54.25 | 60.15 |
| SparseGPT | ✓ | 4:8 | 52.80 | 55.99 | 60.79 | 64.87 | **53.80** | **59.18** | 65.84 | 56.51 | 66.42 |
| Wanda | ✗ | 4:8 | 52.76 | 56.09 | 61.00 | 64.97 | 52.49 | 58.00 | 66.06 | 55.31 | 66.76 |
| BaWA | ✗ | 4:8 | **53.12** | **56.33** | **61.62** | **65.64** | 53.65 | 58.38 | **66.31** | **57.24** | **67.34** |
| Magnitude | ✗ | 2:4 | 44.73 | 48.00 | 53.16 | 61.28 | 45.58 | 49.20 | 59.95 | 49.68 | 60.39 |
| SparseGPT | ✓ | 2:4 | **50.60** | 53.22 | 58.91 | 62.57 | **50.94** | 54.36 | 63.89 | 50.88 | 66.57 |
| Wanda | ✗ | 2:4 | 48.53 | 52.30 | 59.21 | 62.84 | 48.75 | 53.19 | 64.14 | 50.15 | 66.16 |
| BaWA | ✗ | 2:4 | 49.91 | **53.33** | **59.87** | **62.90** | 49.98 | **54.61** | **65.03** | **53.23** | **67.29** |

Table 11: Detailed analysis on zero-shot tasks for LLaMA2-13B.

| Method | Sparsity | BoolQ | RTE | HellaSwag | WinoGrande | ARC-e | ARC-c | OBQA | Avg |
|---|---|---|---|---|---|---|---|---|---|
| Dense | 0% | 80.61 | 65.34 | 60.00 | 72.37 | 79.41 | 48.37 | 35.20 | 63.04 |
| Magnitude | 50% | 57.71 | 55.6 | 54.43 | 65.35 | 70.58 | 38.31 | 27.40 | 52.77 |
| SparseGPT | 50% | **81.38** | **66.06** | 56.09 | **71.51** | 75.20 | 41.72 | 32.00 | 60.57 |
| Wanda | 50% | 81.28 | 60.65 | **57.05** | 70.64 | 75.76 | 42.75 | **32.20** | 60.04 |
| BaWA | 50% | 80.76 | 57.76 | 56.87 | 70.72 | **76.56** | **44.62** | 32.00 | **60.67** |
| Magnitude | 4:8 | 63.39 | 57.76 | 53.97 | 64.72 | 68.52 | 35.84 | 26.00 | 52.89 |
| SparseGPT | 4:8 | **80.58** | **64.62** | 51.94 | **71.98** | 73.70 | **40.87** | 30.60 | **59.18** |
| Wanda | 4:8 | 79.60 | 60.06 | 52.32 | 69.53 | 73.95 | 40.36 | 30.20 | 58.00 |
| BaWA | 4:8 | 80.12 | 62.45 | **53.58** | 68.35 | **74.07** | 38.91 | **31.20** | 58.38 |
| Magnitude | 2:4 | 65.81 | 53.79 | 50.12 | 62.12 | 57.53 | 31.83 | 23.20 | 49.20 |
| SparseGPT | 2:4 | 77.89 | 55.26 | 46.97 | **69.40** | 69.37 | 35.03 | **26.60** | 54.36 |
| Wanda | 2:4 | 75.26 | **56.68** | 46.43 | 66.77 | 68.35 | 34.47 | 24.4 | 53.19 |
| BaWA | 2:4 | **78.26** | 56.32 | **48.5** | 66.93 | **70.79** | **35.49** | 26.00 | **54.61** |

tests on the MMLU dataset (Hendrycks et al., 2021a;b). As shown in Table 13, we compared the performance of various methods on the MMLU dataset. Specifically, we have evaluated the 5-shot performance on the LLaMA-30B model, with unstructured 50%, 4:8 and 2:4 sparsity. The experimental results indicate that, on average, BaWA achieves higher accuracy than existing SOTA methods. This further confirms the effectiveness of BaWA, and effectively demonstrates that pruned models are task-agnostic and generalizable to any downstream task.

## C. Visualization

We have selected several different layers from LLaMA2-7B and analyzed the power factor parameters obtained through BaWA, as detailed in Table 14, 15 and 16. As one can notice in Table 14, the parameters for the weights were consistently close to 1, which is attributed to the absence of outliers in the weight distribution, thus negating the need for a lower power factor to suppress outliers. Additionally, we can note that the parameters for transformer block 2 are closer to 0.5 compared to blocks 15 and 28, which is due to the lower loss in the earlier layers requiring less significant parameter adjustments, which satisfies our observation in Figure 5. Similarly, we have also observed that different operators exhibit distinct tendencies in the selection of activation power factors. For instance, the parameter for Mlp_DOWN is notably higher than other values, which substantiates that the proportion of activation outliers in Mlp_DOWN is lower.

Table 12: Detailed analysis on zero-shot tasks for Mistral-7B.

| Method | Sparsity | BoolQ | RTE | HellaSwag | WinoGrande | ARC-e | ARC-c | OBQA | Avg |
|--------|----------|-------|-----|-----------|------------|-------|-------|------|-----|
| Dense | 0% | 83.43 | 67.51 | 61.25 | 73.80 | 80.83 | 50.43 | 32.80 | 64.30 |
| Magnitude | 50% | 71.13 | 55.96 | **56.63** | 66.30 | 72.60 | 41.30 | 27.20 | 55.87 |
| SparseGPT | 50% | 83.06 | 58.12 | 55.82 | **72.06** | **75.59** | 42.75 | 28.00 | 59.34 |
| Wanda | 50% | 82.91 | 58.12 | 55.18 | 71.27 | 74.66 | 43.17 | 27.20 | 58.93 |
| BaWA | 50% | **83.21** | 63.54 | 56.17 | 70.56 | 75.42 | **43.26** | **29.00** | **60.17** |
| Magnitude | 4:8 | 74.37 | 56.68 | **53.72** | 65.11 | 68.69 | 36.95 | 24.20 | 54.25 |
| SparseGPT | 4:8 | 79.11 | 60.65 | 51.64 | **68.67** | **71.93** | **37.97** | 25.60 | 56.51 |
| Wanda | 4:8 | 75.32 | 63.90 | 50.32 | 67.80 | 70.41 | 35.84 | 23.60 | 55.31 |
| BaWA | 4:8 | **81.25** | **65.70** | 52.36 | 68.27 | 71.80 | 36.69 | **24.60** | **57.24** |
| Magnitude | 2:4 | 62.65 | 55.60 | **48.74** | 61.88 | 66.75 | 31.14 | 21.00 | 49.68 |
| SparseGPT | 2:4 | 71.41 | 54.15 | 46.61 | 64.48 | 66.62 | 32.08 | 20.80 | 50.88 |
| Wanda | 2:4 | 67.80 | 57.76 | 44.61 | 63.93 | 64.98 | 30.97 | 21.00 | 50.15 |
| BaWA | 2:4 | **74.16** | **60.65** | 46.78 | **66.61** | **67.00** | **33.02** | **24.40** | **53.23** |

Table 13: MMLU performance of LLaMA-30B with unstructured 50%, 4:8 and 2:4 sparsity.

| Method | Sparsity | Humanities | Other | Social_Sciences | Stem | Avg |
|--------|----------|-----------|-------|-----------------|------|-----|
| Dense | 0% | 56.00 | 64.31 | 67.47 | 47.00 | 58.33 |
| Magnitude | 50% | 38.02 | 47.12 | 47.22 | 35.49 | 41.48 |
| Wanda | 50% | 48.65 | **59.16** | 59.57 | 42.63 | 52.02 |
| SparseGPT | 50% | **50.69** | 57.68 | 59.18 | 41.58 | 52.05 |
| BaWA | 50% | 49.27 | 58.77 | **61.16** | **43.64** | **52.71** |
| Magnitude | 4:8 | 27.61 | 32.89 | 33.77 | 31.27 | 30.95 |
| Wanda | 4:8 | 43.57 | 54.20 | 53.33 | **40.72** | 47.42 |
| SparseGPT | 4:8 | **44.82** | **54.29** | 54.18 | 39.15 | 47.70 |
| BaWA | 4:8 | 44.06 | 54.23 | **55.05** | 39.77 | **47.76** |
| Magnitude | 2:4 | 28.84 | 34.86 | 35.72 | 31.11 | 32.19 |
| Wanda | 2:4 | 38.30 | 43.39 | 45.95 | 36.28 | 40.65 |
| SparseGPT | 2:4 | 39.68 | **44.60** | 46.52 | 36.23 | 41.49 |
| BaWA | 2:4 | **41.82** | 44.35 | **47.25** | **36.54** | **41.82** |

Table 14: Visualization of the parameters acquired for different transformer blocks on LLaMA2-7B with unstructured 50% sparsity.

| Operator | Transformer Block-2 | | | Transformer Block-15 | | | Transformer Block-28 | | |
|----------|------------|------------|------------|------------|------------|------------|------------|------------|------------|
| | $\theta_1$ | $\theta_2$ | $\theta_3$ | $\theta_1$ | $\theta_2$ | $\theta_3$ | $\theta_1$ | $\theta_2$ | $\theta_3$ |
| Attn_Q | 1.00 | 1.00 | 0.43 | 1.00 | 1.00 | 0.32 | 0.99 | 1.00 | 0.38 |
| Attn_K | 0.98 | 0.97 | 0.44 | 0.98 | 0.99 | 0.38 | 0.98 | 1.00 | 0.34 |
| Attn_V | 0.97 | 1.12 | 0.37 | 1.02 | 1.06 | 0.35 | 1.02 | 1.08 | 0.33 |
| Attn_O | 1.01 | 0.97 | 0.43 | 1.02 | 0.99 | 0.37 | 1.00 | 0.98 | 0.35 |
| Mlp_GATE | 0.98 | 1.01 | 0.43 | 1.02 | 0.99 | 0.37 | 1.00 | 0.98 | 0.35 |
| Mlp_UP | 0.95 | 0.95 | 0.32 | 1.00 | 1.00 | 0.35 | 1.01 | 0.96 | 0.38 |
| Mlp_DOWN | 1.04 | 1.06 | 0.54 | 1.00 | 0.99 | 0.48 | 0.99 | 0.98 | 0.45 |

Table 15: Visualization of the parameters acquired for different transformer blocks on LLaMA2-7B with semi-structured 4:8 sparsity.

| Operator | Transformer Block-2 | | | Transformer Block-15 | | | Transformer Block-28 | | |
|---|---|---|---|---|---|---|---|---|---|
| | $\theta_1$ | $\theta_2$ | $\theta_3$ | $\theta_1$ | $\theta_2$ | $\theta_3$ | $\theta_1$ | $\theta_2$ | $\theta_3$ |
| Attn_Q | 0.52 | 0.48 | 0.41 | 0.54 | 0.43 | 0.30 | 0.52 | 0.48 | 0.36 |
| Attn_K | 0.49 | 0.41 | 0.45 | 0.56 | 0.61 | 0.37 | 0.48 | 0.49 | 0.32 |
| Attn_V | 0.53 | 0.59 | 0.36 | 0.52 | 0.49 | 0.38 | 0.54 | 0.58 | 0.32 |
| Attn_O | 0.59 | 0.51 | 0.41 | 0.53 | 0.52 | 0.36 | 0.52 | 0.72 | 0.34 |
| Mlp_GATE | 0.55 | 0.53 | 0.42 | 0.43 | 0.52 | 0.36 | 0.48 | 0.62 | 0.36 |
| Mlp_UP | 0.51 | 0.49 | 0.29 | 0.49 | 0.56 | 0.38 | 0.49 | 0.55 | 0.39 |
| Mlp_DOWN | 0.41 | 0.54 | 0.51 | 0.52 | 0.58 | 0.41 | 0.56 | 0.61 | 0.42 |

Table 16: Visualization of the parameters acquired for different transformer blocks on LLaMA2-7B with semi-structured 2:4 sparsity.

| Operator | Transformer Block-2 | | | Transformer Block-15 | | | Transformer Block-28 | | |
|---|---|---|---|---|---|---|---|---|---|
| | $\theta_1$ | $\theta_2$ | $\theta_3$ | $\theta_1$ | $\theta_2$ | $\theta_3$ | $\theta_1$ | $\theta_2$ | $\theta_3$ |
| Attn_Q | 0.42 | 0.45 | 0.41 | 0.34 | 0.58 | 0.25 | 0.48 | 0.41 | 0.34 |
| Attn_K | 0.41 | 0.46 | 0.40 | 0.51 | 0.56 | 0.32 | 0.45 | 0.44 | 0.32 |
| Attn_V | 0.51 | 0.51 | 0.36 | 0.43 | 0.62 | 0.31 | 0.51 | 0.50 | 0.31 |
| Attn_O | 0.53 | 0.58 | 0.40 | 0.42 | 0.52 | 0.34 | 0.47 | 0.51 | 0.32 |
| Mlp_GATE | 0.51 | 0.51 | 0.38 | 0.46 | 0.53 | 0.36 | 0.42 | 0.51 | 0.25 |
| Mlp_UP | 0.42 | 0.46 | 0.28 | 0.44 | 0.54 | 0.32 | 0.43 | 0.51 | 0.31 |
| Mlp_DOWN | 0.39 | 0.51 | 0.50 | 0.48 | 0.51 | 0.40 | 0.51 | 0.52 | 0.38 |

We then compare the power factors acquired from different sparsity patterns, as indicated in Table 14, 15, and 16. It is noteworthy that when we adopt semi-structured N:M sparsity, the power factors for weight can be greatly reduced. This is due to the more similar distributions among adjacent channels, as revealed in Figure 1. Furthermore, the power factor for activation should be lower when adopting BaWA on N:M sparsity.

## D. Robustness Analysis

In this part, we perform a robustness analysis for different pruning methods. Here, our focus is primarily on the pruning process of BaWA, rather than the parameter search process. As shown in Table 17, we report the mean and standard deviation under 5 random seeds for sparse LLaMA1 and LLaMA2 family models at 50% and 60% sparsity. As one can

Table 17: WikiText-2 perplexity performance of BaWA for sparse LLaMA-1 and LLaMA2 family models at 50% and 60% sparsity. We report the mean and standard deviation under 5 random seeds. Note that, when the standard deviation is less than 0.01, we denote it as 0.00, with the sign *. However, the actual standard deviation is not equal to zero.

| Method | Sparsity | LLaMA-1 | | | LLaMA2 | |
|---|---|---|---|---|---|---|
| | | 7B | 13B | 30B | 7B | 13B |
| Dense | 0% | 5.68 ($\pm$0.00) | 5.09 ($\pm$0.00) | 4.77 ($\pm$0.00) | 5.12 ($\pm$0.00) | 4.57 ($\pm$0.00) |
| SparseGPT | 50% | 7.26 ($\pm$0.04) | 6.24 ($\pm$0.04) | 5.33 ($\pm$0.03) | 6.52 ($\pm$0.04) | 5.63 ($\pm$0.01) |
| Wanda | 50% | 7.25 ($\pm$0.03) | 6.15 ($\pm$0.01) | 5.24 ($\pm$0.03) | 6.44 ($\pm$0.01) | 5.57 ($\pm$0.03) |
| BaWA | 50% | 7.05 ($\pm$0.03) | 5.95 ($\pm$0.01) | 5.01 ($\pm$0.01) | **6.23*** ($\pm$0.00) | **5.37*** ($\pm$0.00) |
| SparseGPT | 60% | 10.34 ($\pm$0.14) | 8.44 ($\pm$0.15) | 6.69 ($\pm$0.03) | 0.58 ($\pm$0.11) | 7.71 ($\pm$0.11) |
| Wanda | 60% | 10.62 ($\pm$0.05) | 8.72 ($\pm$0.03) | 6.54 ($\pm$0.01) | 9.91 ($\pm$0.01) | 7.87 ($\pm$0.07) |
| BaWA | 60% | 9.80 ($\pm$0.02) | 7.60($\pm$0.01) | 6.15 ($\pm$0.01) | 8.78 ($\pm$0.01) | 6.80($\pm$0.01) |

**Algorithm 1** PyTorch code for the pruning process of BaWA

```
# W: weight matrix (C_out, C_in);
# X: input matrix (N * L, C_in);
# Θ: learned parameters for BaWA;
# s: desired sparsity level, between 0 and 1;
def prune(W, X, Θ, s):
  θ₁,θ₂,θ₃ = Θ
  w_norm = weight_normalization(W, θ₁,θ₂)
  act_metric = X.norm(p=2, dim=0) ** θ₃
  metric = w_norm.abs() * act_metric

  _, sorted_idx = torch.sort(metric, dim=1)
  pruned_idx = sorted_idx[:,:int(C_in * s)]
  W.scatter_(dim=1, index=pruned_idx, src=0)
  return W

def weight_normalization(W, θ₁, θ₂):
  W_row_norm = W.norm(p=2, dim=-1)
  W_col_norm = W.norm(p=2, dim=0)
  row_metric = (1/W_row_norm) ** θ₁
  col_metric = (1/W_col_norm) ** θ₂
  return W * (row_metric + col_metric)
```

**Algorithm 2** Pytorch code for the proposed zeroth-order optimization

```
# layer: the specific transformer block that
    needs to be optimized;
# inputs: the sampled input data for
    optimization;
# fouts: the output data of dense transformer
    block;
# Θ: learned parameters for BaWA;

def zo_optimizer(layer, inputs, fouts, Θ):
  z = torch.normal(mean=0, std=1)
  Θ = zo_update(Θ, 1, ε, z)
  loss1 = lossfunc(layer, inputs, fouts)
  Θ = zo_update(Θ, -2, ε, z)
  loss2 = lossfunc(layer, inputs, fouts)
  Θ = zo_update(Θ, 1, ε, z)
  grad = ((loss1 - loss2) / 2
  Θ = zo_update(Θ, lr, grad, z)

def zo_update(Θ, lr, grad, z):
  for θ in Θ:
    θ -= lr * grad * z
  return Θ
```

notice, compared with previous methods, the standard deviation is lower, indicating its robustness under different random seeds.

# E. Code Details

In this section, we further present the code details for implementing BaWA, enabling readers to easily replicate the experimental results reported. The code for BaWA's pruning process and parameter optimization is shown in Algorithm 1 and 2, respectively.

The pruning process of BaWA can be implemented and integrated seamlessly within a single forward pass of the LLM model, can presented in Algorithm 1. Given a pre-trained LLM, we compute our pruning metric from the initial to the final layers of the network. After pruning a preceding layer, the subsequent layer receives updated input activations, based on which its pruning metrics will be computed. As one can notice, the complexity of the pruning process of BaWA is $O(d_{hidden}^2)$, which is the same as Wanda. $d_{hidden}$ denotes the channel dimension of the hidden layer.

However, we also introduce a parameter optimization stage in BaWA, which uses a zeroth-order optimizer to adjust the corresponding power factors in Equation (6). As Algorithm 2 shows, the entire parameter optimization process requires two times loss computations and four times parameter updates. Each loss computation requires one forward pass. Therefore, the complexity of the whole parameter optimization process is consistent with the complexity of the model inference process, which is $O(BQ \cdot d_{hidden}^3)$. Here, $B$ denotes the batch size and $Q$ denotes the sequence length.

# F. Weight Reconstruction with Efficient Pruning Mask

As mentioned in Section 2, the pruning process can be divided into two stages, including pruning mask selection and weight reconstruction. Our evaluation demonstrates that BaWA presents a high-performance pruning mask metric, which can preserve the performance of original LLMs without introducing any weight update. It is trivial to combine BaWA with other weight reconstruction methods to achieve better pruning performance. Consequently, we adopt two novel weight reconstruction methods, DS⊘T (Zhang et al., 2023) and ADMM-Iter (Boža, 2024). DS⊘T introduces a novel weight connection reconstruction strategy that iteratively grows and prunes weight connection based on their proposed metric. ADMM-Iter views the weight update process as an ADMM problem and solves it by iteratively adjusting the weight data that are not masked. We use two different pruning metrics, Wanda and BaWA, and compare their pruning results, as shown in Table 18.

Table 18: Weight reconstruction strategies with efficient pruning mask achieve better performance.

| Method | Sparsity | LLaMA | | | | LLaMA2 | | |
|---|---|---|---|---|---|---|---|---|
| | | 7B | 13B | 30B | 65B | 7B | 13B | 70B |
| Wanda w/ DS⊘T | 50% | 7.12 | 6.08 | 5.12 | 4.54 | 6.31 | 5.48 | 3.95 |
| BaWA w/ DS⊘T | 50% | 6.97 | 5.94 | 5.01 | 4.37 | 6.28 | 5.42 | 3.87 |
| Wanda w/ ADMM-Iter | 50% | 7.06 | 6.07 | 5.18 | 4.51 | 6.33 | 5.52 | 3.95 |
| BaWA w/ ADMM-Iter | 50% | 6.91 | 5.93 | 4.98 | 4.37 | 6.22 | 5.33 | 3.81 |
| Wanda w/ DS⊘T | 4:8 | 8.45 | 7.25 | 5.91 | 5.26 | 7.83 | 6.47 | 4.43 |
| BaWA w/ DS⊘T | 4:8 | 8.01 | 6.83 | 5.65 | 4.93 | 7.36 | 6.19 | 4.35 |
| Wanda w/ ADMM-Iter | 4:8 | 8.13 | 6.97 | 5.73 | 5.17 | 7.71 | 6.41 | 4.40 |
| BaWA w/ ADMM-Iter | 4:8 | 7.96 | 6.76 | 5.63 | 4.95 | 7.34 | 6.11 | 4.31 |
| Wanda w/ DS⊘T | 2:4 | 10.89 | 9.05 | 6.76 | 6.14 | 10.46 | 8.09 | 5.11 |
| BaWA w/ DS⊘T | 2:4 | 10.21 | 7.91 | 6.42 | 5.69 | 9.84 | 7.08 | 4.86 |
| Wanda w/ ADMM-Iter | 2:4 | 9.90 | 8.47 | 6.72 | 6.23 | 9.89 | 7.94 | 5.19 |
| BaWA w/ ADMM-Iter | 2:4 | 9.71 | 7.86 | 6.39 | 5.60 | 9.75 | 7.04 | 4.71 |

It is worth noting that using BaWA as the pruning metric achieves superior LLM performance compared to the adoption of Wanda. Furthermore, the performance of pruned LLMs is enhanced with the addition of a weight reconstruction strategy beyond what is achieved by pruning mask alone in most cases. This underscores the complementary nature of the two phases: an effective pruning mask can significantly improve pruning performance, while an appropriate weight reconstruction strategy can further refine the results in post-training pruning.

