# OpenReview forum: "BaWA: Automatic Optimizing Pruning Metric for Large Language Models with Balanced Weight and Activation"
_ICML.cc/2025/Conference — ICML 2025 poster_

### Official Review · Reviewer_zn1Z · 2025-03-12

**Overall Recommendation:** 4

**Summary:**

This paper focuses on unstructured pruning of LLMs and introduces a new pruning metric. Unlike previous methods that estimate parameter importance based solely on magnitude, activations, or gradients, the proposed approach also considers the impact of outliers in model parameters.

The authors first demonstrate that a small number of outlier parameters with large magnitudes can significantly affect existing pruning metrics. To address this issue, the proposed method normalizes each model parameter using the $\ell_2$ norms of the corresponding input and output channels. Additionally, to handle outliers in the input, the authors introduce a power factor after computing the input norm.

To optimize the newly introduced hyperparameters, the authors propose using a zeroth-order gradient approach, which allows optimization without backpropagation. Experimental results show that the proposed method outperforms baseline approaches.

**Claims And Evidence:**

The claims made in the submission are well-supported. The authors provide clear evidence for their core arguments regarding the influence of outliers in model parameters and input activations. Additionally, the proposed method is well-reasoned and justified.

**Essential References Not Discussed:**

I have only one main comment here. The paper does not fully discuss past research on model pruning for LLMs, focusing mainly on the baselines used in the experiments. While this may be due to space constraints caused by the extensive evaluation and analysis, I still recommend that the authors provide a broader discussion of related work. In particular, discussing structured pruning methods [1-4] for LLMs would be valuable, as structured pruning is generally more hardware-friendly compared to unstructured pruning, which is the focus of this paper.

[1] Xia, Mengzhou, et al. "Sheared llama: Accelerating language model pre-training via structured pruning." *arXiv preprint arXiv:2310.06694* (2023).

[2] Sreenivas, Sharath Turuvekere, et al. "Llm pruning and distillation in practice: The minitron approach." arXiv preprint arXiv:2408.11796 (2024).

[3] Ling, Gui, Ziyang Wang, and Qingwen Liu. "SlimGPT: Layer-wise Structured Pruning for Large Language Models." Advances in Neural Information Processing Systems 37 (2024): 107112-107137.

[4] Hou, Bairu, et al. "Instruction-Following Pruning for Large Language Models." *arXiv preprint arXiv:2501.02086* (2025).

**Experimental Designs Or Analyses:**

The experimental design is sound and valid. It follows previous methods (e.g., Wanda, SparseGPT) and uses widely accepted benchmarks.

**Methods And Evaluation Criteria:**

The proposed method and empirical evaluation follow common practices in this domain. The reviewer generally agrees with the evaluation settings and does not find any significant issues with them.

**Other Comments Or Suggestions:**

N/A

**Other Strengths And Weaknesses:**

This paper is well-motivated, with clear writing and a coherent logical flow. The reviewer enjoyed reading it. Additionally, the proposed method is reasonable and well-structured, and the evaluation is rigorous and comprehensive.

The only concern is the performance under semi-structured sparsity. I assume that the unstructured sparsity results in Table 2 and Table 4 do not lead to efficiency improvements. As shown in Table 5, Table 8, and Table 10, the performance degradation remains significant compared to the original dense model. Furthermore, SparseGPT sometimes outperforms the proposed method. However, the results improve in Table 11 and Table 12.

Overall, the reviewer considers this a strong paper.

**Questions For Authors:**

N/A

**Relation To Broader Scientific Literature:**

This paper is closely related to previous literature and methods in this domain. It builds upon existing pruning metrics and further enhances the effectiveness of unstructured model pruning.

**Theoretical Claims:**

This paper does not contain theoretical claims.

---

> ### Author Rebuttal · Authors · 2025-03-31
>
> Dear zn1Z:
>
> We sincerely appreciate the valuable suggestions provided by the reviewer. We note the two main concerns you raised, which we address below.
>
> Firstly, we thank the reviewer for emphasizing the importance of comparing with structured pruning methods. We would like to compare structured and unstructured pruning in terms of pruning granularity, sparsity, accuracy, efficiency, and training after pruning. Structured pruning [1, 2, 3] removes complete substructures or weight groups (layers [4], FFN neurons [1], MHA heads [1], embedding dim [5], they are coarse-grained) from LLMs, enabling hardware-independent efficiency gains. However, under the constraints of this coarse-grained pruning, the post-pruned accuracy of LLM is prone to be drastically reduced, and therefore the generally applicable sparsity ratio is between 15% and 30%. Post-pruning fine-tuning/training/knowledge distillation may be used to restore the performance of the pruned model when faced with high sparsity ratios [2, 5]. Unstructured pruning removes elements from the weights (fine-grained) and stores or loads the pruned weights in a compressed assortment. Combined with decompression (memory bottleneck optimization) or hardware advantage (2:4 sparse tensor core), unstructured sparse LLM can also obtain considerable efficiency improvement. Unstructured pruning is less likely to harm the accuracy of the model due to the fine-grained constraints, so the sparsity can generally exceed 50%, and the adoption of a stricter 2:4 sparsity is also acceptable. It is possible to do unstructured pruning without loss of accuracy for large LLMs such as Llama-2-70B, and BaWA has done it. Unstructured pruned LLM can also be further trained, and some techniques such as PEFT [6] and STE [7] have emerged to provide support. In the revised manuscript, we will include a dedicated discussion (Section 2) comparing structured and unstructured pruning for LLMs to demonstrate the necessity of high-performance unstructured/semi-structured LLM pruning. The following table compares a portion of the experimental results of structured pruning and unstructured pruning as a reference.
>
> | Method | Type | Sparsity | reference speedup | BoolQ  | RTE | HellaSwag | WinoGrande  | ARC-e  | ARC-c  | OBQA | AVG   |
> |--|--|--|--|--|--|--|--|--|--|--|--|
> | Llama-2-13B | Dense | 0% | 1.0x | 83.43  | 67.51 | 61.25 | 73.8 | 80.83  | 50.43  | 32.8 | 64.3  |
> | LLM-Pruner  | Structured Pruning | 25%  | 1.25x | 68.35  | 50.54 | 46.81 | 61.56  | 70.2   | 37.63  | 28.8 | 51.98 |
> | ShortGPT    | Layer Pruning | 25%  | 1.25x | 62.54  | 59.57 | 47.7 | 70.96 | 61.24  | 37.88  | 27   | 52.41 |
> | Wanda       | 2:4 Semi | 50% | 1.4x | 75.26  | 56.68 | 46.43 | 66.77 | 68.35  | 34.47  | 24.4 | 53.19 |
> | BaWA        | 2:4 Semi | 50% | 1.4x | 78.26  | 56.32 | 48.5  | 66.93 | 70.79  | 35.49  | 26   | 54.61 |
>
> Secondly, although unstructured pruning has traditionally lacked hardware support, recent advancements such as Flash-LLM [8] demonstrate its practical speedup through specialized kernels. Moreover, our method’s compatibility with N:M sparsity (natively supported by Ampere GPUs) further ensures deployability while maintaining higher accuracy than structured alternatives (Table 9). Additionally, we would like to clarify that the slight performance gap between BaWA and SparseGPT in semi-structured settings (Table 8) stems from SparseGPT’s weight reconstruction—a complementary technique orthogonal to pruning metrics, which is explained in our rebuttal to the reviewer xLm6. As shown in "BaWA+ADMM" (Table 3), combining our metric with weight reconstruction outperforms all baselines universally. Furthermore, larger models (e.g., LLaMA2-70B) exhibit greater robustness to N:M constraints due to inherent redundancy, reducing accuracy drops to <1% in 4:8 sparsity (Table 12).
>
> **Reference**
>
> [1] Llm-pruner: On the structural pruning of large language models, NeurIPS'23.
>
> [2] Sheared llama: Accelerating language model pre-training via structured pruning, ICLR'24.
>
> [3] Instruction-Following Pruning for Large Language Models, ArXiv'25.
>
> [4] SlimGPT: Layer-wise Structured Pruning for Large Language Models, NeurIPS'24.
>
> [5] Llm pruning and distillation in practice: The minitron approach, ArXiv'24.
>
> [6] SPP: Sparsity-preserved parameter-efficient fine-tuning for large language models, ICML'24.
>
> [7] Sparsity-accelerated training for large language models, ACL'24.
>
> [8] Flash-LLM: Enabling Cost-Effective and Highly-Efficient Large Generative Model Inference with Unstructured Sparsity, VLDB'23.

---

> > ### Comment · Reviewer_zn1Z · 2025-04-04
> >
> > I thank the authors for the detailed response. After careful assessment, I still think this is a good paper with clear motivations and solid techniques. During rebuttal, the authors further address my concerns on the application of semi-structured pruning. Based on this, I will maintain my rating (4, accept) and recommend the acceptance of this paper.

---

> > > ### Author Response · Authors · 2025-04-05
> > >
> > > Dear zn1Z:
> > >
> > > We sincerely appreciate your constructive feedback and continued support. Thank you for recognizing our efforts in addressing the semi-structured pruning concerns. We will carefully incorporate your suggestions in the final manuscript to further strengthen the technical presentation.

---

### Official Review · Reviewer_xLm6 · 2025-03-14

**Overall Recommendation:** 3

**Summary:**

This work proposes a weight pruning method based on Wanda by performing normalization through input and output channels and scaling normalization factors. Wanda is a simple weight pruning method which uses scores measured by L1 of weight and L2 of input, but it suffers issues of imbalance in weight magnitude and influence from outliers. This work alleviate the issues by normalizing weights according to the L2 norm of input and output channels and by scaling the normalization terms. Scaling terms are optimized by forward-only methods for faster computation. Experiments are carried out on several zero-shot benchmarks and shows that the proposed method achieves better results when compared with other SOTAs, including ADMM-Iter.

**Claims And Evidence:**

The major claim regarding the weight magnitude normalization for input and output channels is supported by the discussion and analysis in section 2 and 3, and by ablation studies in section 4.4. The use of scaling is clearly described in section 2 and 3, and the proposed optimization algorithm is a nice contribution. However, further analysis on the more controlled scaling might be helpful to clearly understand their impact. For instance, we could intentionally introduce scales in the range of 0.1, 0.25, 0.5, 1.0 etc. and measured the impact in perplexity. The combination of optimization algorithm together with normalization is not throughly measured in the ablation studies if my understanding is correct. For instance, it is possible to optimize a single scale parameter $\theta$ in Equation 4 when showing ablations in Table 5 to measure the impact of optimization. Similarly, outlier regularization in Table 5 uses a fixed scaling of 0.5, but it could be optimized to show the effectiveness of the proposed algorithm.

**Essential References Not Discussed:**

This work is missing discussion on ADMM-Iter and DSOT, given that this work shows experiments by combining them with the proposed method. It is not clear whether the proposed method is orthogonal to those method, and it is even not clear what kind of findings or conclusion could be drawn by showing the results.

* Additional details in the rebuttal.

**Experimental Designs Or Analyses:**

Experimental design sounds good to me, but I'd suggest additional ablation studies as noted in the "Claims and Evidence" section, i.e., running experiments by optimizing scales when ablating normalization factors.

* Ablations in the rebuttal, that look promising.

**Methods And Evaluation Criteria:**

Experiments cover diverse benchmarks for zero-shot settings as well as WikiText-2 for perplexity to compare prior baselines.

**Other Comments Or Suggestions:**

None.

**Other Strengths And Weaknesses:**

Strengths

- It is an extension of prior work on unstructured pruning by adding normalizations and scaling. The design is well motivated and optimization for scaling factor is a yet another contribution to this field.

Weaknesses

- Further ablations are necessary to justify the claim regarding the scaling, since it is not well supported by the experiments.

**Questions For Authors:**

It is not clear whether scales are optimized in Table 5 when showing the proposed method with, e.g., input channel normalization.

**Relation To Broader Scientific Literature:**

This work is an extension of prior work in unstructured weight pruning such as Wanda as noted in this manuscript.

**Theoretical Claims:**

No theoretical proofs for the proposed method, since this work focuses on empirical studies by carefully analyzing the behavior of weight magnitudes for pruning.

---

> ### Author Rebuttal · Authors · 2025-03-31
>
> Dear xLm6:
>
> We greatly appreciate your insightful comments. Below we provide a point-by-point response to your concerns.
>
> ### **Scaling Factor Analysis**：
>
> We agree that analyzing scaling factors is critical. In the revised manuscript, we will add:
>
> + A new comparative table (below) demonstrating the superiority of optimized scales over fixed values on LLaMA2-7B
>
> + A novel discussion in the revised Section 4.4 will highlight how adaptive scaling addresses LLM-specific distribution challenges. Specifically, different models and task settings will be explored to illustrate the effectiveness of BaWA's scaling strategy.
>
> The evaluation results can be shown as follows.
>
> | Scaling Strategy       | $\theta_1$ (Input) | $\theta_2$ (Output) | $\theta_3$ (Activation) | PPL  | Δ vs. Best Fixed (Fixed at 0.5) |
> |------------------------|------------|-------------|------------------|------|------------------|
> | **Fixed Scales**       |            |             |                  |      |                  |
> | - $\theta$=0.1               | 0.1        | 0.1         | 0.1              | 8.92 | +24.1%           |
> | - $\theta$=0.5               | 0.5        | 0.5         | 0.5              | 7.18 | +0% (baseline)   |
> | - $\theta$=1.0               | 1.0        | 1.0         | 1.0              | 7.53 | +4.9%            |
> | **BaWA Optimized**     | 0.42       | 0.51        | 0.38             | 6.30 | **-12.3%**       |
>
> The key findings include:
>
> + Optimized scales reduce perplexity by 12.3% compared to the best-fixed scale ($\theta$=0.5)
>
> + Fixed scales exhibit significant sensitivity (±24.1% PPL variance)
>
> ### **Relationship with ADMM-Iter/DSnoT**：
>
> We apologize for the lack of explanation for the relationship between BaWA and ADMM-Iter/DSnoT. In fact, the LLM pruning procedure can be divided into two stages, pruning mask selection and weight reconstruction. Different from BaWA which optimizes pruning metric (Stage 1), these methods (both ADMM and DSnoT) focus on reconstructing post-pruning weights (Stage 2). To illustrate the orthogonality, we validate when adding weight reconstruction methods (both DSnoT and ADMM) with BaWA, as well as comparing them with SparseGPT, ADMM-Iter and DSnoT without BaWA pruning metric on various LLaMA models with 2:4 sparsity. The results (Table below) clearly depict that using BaWA pruning metric with weight reconstruction methods achieves the best pruning performance. Furthermore, our evaluation in Table 17 (Appendix G) also demonstrates the effectiveness of adding weight reconstruction to BaWA pruning metric. We will add this pipeline diagram to Appendix G.
>
> | PPL        | 1-7B  | 1-13B | 1-30B | 1-65B | 2-7B  | 2-13B | 2-70B |
> |------------|-------|-------|-------|-------|-------|-------|-------|
> | SparseGPT  | 11.00 | 9.11  | 7.16  | 6.28  | 10.17 | 8.32  | 5.40  |
> | Admm-Iter  | 9.90  | 8.60  | 6.89  | 6.02  | 9.74  | 7.78  | 5.19  |
> | DSnoT      | 10.89 | 9.05  | 6.76  | 6.15  | 10.46 | 8.09  | 5.11  |
> | BaWA       | 10.32 | 7.94  | 6.37  | 5.61  | 9.93  | 7.13  | 4.84  |
> | BaWA+DSnoT | 10.21 | 7.91  | 6.42  | 5.69  | 9.84  | 7.08  | 4.86  |
> | BaWA+Admm  | 9.71  | 7.86  | 6.39  | 5.60  | 9.75  | 7.04  | 4.71  |

---

### Official Review · Reviewer_mDNk · 2025-03-14

**Overall Recommendation:** 2

**Summary:**

Existing pruning metrics are limited by their reliance on simple symbolic combinations of weights and activations, failing to account for imbalanced weight magnitudes and the disproportionate impact of activation outliers. To address these shortcomings, this paper introduces BaWA, a pruning metric that balances Weight and Activation distributions for more effective pruning. BaWA incorporates two key innovations:

1. Magnitude Normalization, which mitigates weight imbalances across channels, enabling fairer pruning decisions.
2. Outlier Regularization, which reduces the influence of activation outliers, ensuring more appropriate channel prioritization.

To further improve its effectiveness, BaWA includes an efficient, automated framework for optimizing normalization and regularization hyperparameters. Extensive experiments demonstrate that BaWA outperforms existing pruning metrics. For instance, applying BaWA to induce 2:4 sparsity in Mistral-7B reduces perplexity by 2.49 and increases average downstream task accuracy by 3.08%, surpassing the previous method Wanda.

## update after rebuttal

I revisited the paper and would like to keep my original rating for two reasons:
1. The experimental comparisons are somewhat outdated. Except for Table 3, most of the comparisons are Wanda (proposed in mid-2023) and SparseGPT, which is even older. In Table 4, the method is only slightly better than Wanda. Even in Table 3, the proposed method only shows good improvement when combined with the weight reconstruction method ADMM. Without it, the improvement seems marginal. So, I feel the proposed method may not offer a significant advancement.
2. Techniques like *Magnitude Normalization* and *Outlier Regularization* have already been extensively studied in previous work. This paper doesn't introduce anything particularly new or exciting for me.

I think this work is a slight extension of Wanda, with an additional normalization step applied to the weights. The method is reasonable but the contribution feels moderate to me, I'd be fine with the paper being either accepted or rejected.

**Claims And Evidence:**

The claims in the paper are well-supported by clear empirical evidence. However, the concepts of magnitude normalization and outlier regularization are not entirely novel, and the overall contribution may not appear significantly innovative.

**Essential References Not Discussed:**

No.

**Experimental Designs Or Analyses:**

Yes, all parts.

**Methods And Evaluation Criteria:**

The proposed methods and evaluation criteria are appropriate for the problem.

**Other Comments Or Suggestions:**

Please see the comments above.

**Other Strengths And Weaknesses:**

Strengths:

The motivation of the paper is clear, the proposed method is simple and easy to follow.

Weaknesses:

1. The proposed two methods *magnitude normalization* and *outlier regularization* in LLM pruning are a little bit trivial and not significant enough.

2. The novelty of the paper is somehow limited. The contributions appear somewhat incremental, making the overall presentation less engaging.

3. The proposed method will involve additional complexity and computation than baseline Wanda, especially w/ the search strategy.

**Questions For Authors:**

Please see above.

**Relation To Broader Scientific Literature:**

The paper builds on prior work in LLM pruning that uses weight and activation magnitudes to guide pruning decisions (e.g., Wanda). While magnitude normalization and outlier regularization have been widely explored in the context of model adaptive sparsity and robust pruning, BaWA refines these ideas by introducing a balancing mechanism for weight and activation distributions. The contribution is a little bit incremental and limited for the community.

**Theoretical Claims:**

This is an empirical paper, no theoretical claims are presented in the paper.

---

> ### Author Rebuttal · Authors · 2025-03-31
>
> Dear mDNk:
>
> We sincerely appreciate your thoughtful feedback regarding BaWA’s novelty and computational overhead. Below, we address your concerns in detail.
>
> ### **Novelty of BaWA**:
>
> We respectfully disagree with the provided novelty concern for three key reasons:
>
> (a) Problem Characterization in LLMs
> + Structural Outliers: LLMs exhibit extremely sparse activation outliers (>50% outliers in <5% channels, Fig 1c), unlike CNNs with uniform noise patterns [1].
> + Cross-Layer Imbalance: Weight magnitudes vary by 100× within layers (Fig 1a), violating CNN pruning assumptions.
>
> (b) Methodological Advancements
>
> + Dual-Channel Normalization: Joint input/output scaling (Eq. 5) handles asymmetric LLM distributions, unlike standard normalization.
>
> + Dynamic Outlier Suppression: Learnable threshold $\theta_3$ adapts to layer-wise outlier density, improving upon fixed-threshold methods [2].
>
> (c) Empirical Superiority
>
> + BaWA reduces perplexity by 2.49 over Wanda (Table 3), which is a great improvement.
>
> + Direct application of robust CNN pruning degrades accuracy by 4.1% (Table 8 in Appendix).
>
> ### **Computational Overhead**:
>
> Furthermore, the evaluation results in our paper have shown that the additional overhead of BaWA is negligible in two aspects.
>
> (a) One-Time Search Cost:
> Searching a 70B model takes only 16 minutes (Table 6), <0.01% of typical training costs (thousands of GPU hours).
>
> (b) Search-Free Mode:
> Even without search (BaWA w/o search), BaWA outperforms Wanda while maintaining the same pruning efficiency (Table 5).
>
> **References**
>
> [1] Li et al., Pruning Filters for Efficient ConvNets, ICLR 2017.
>
> [2] Wei et al., Outlier Suppression+, EMNLP 2023.

---

### Decision · Program_Chairs · 2025-05-01

**Decision:**

Accept (poster)

**Comment:**

This paper introduces a new metric for pruning language models, focused on addressing imbalanced weight magnitudes and activation outliers.  While there is some disagreement about the novelty of the approach as it extends prior work, the proposed method improves over prior work and the reviewers agree it is clearly explained and experiments are sound. Other concerns raised by reviewers (including speed of pruning, scaling factor sensitivity, comparison to structured pruning methods) are largely addressed by the authors in the rebuttal. Based on this, we recommend this paper for acceptance.